# Molecular Characterization of a *Dirofilaria immitis* Cysteine Protease Inhibitor (Cystatin) and Its Possible Role in Filarial Immune Evasion

**DOI:** 10.3390/genes10040300

**Published:** 2019-04-12

**Authors:** Xiaowei Dong, Jing Xu, Hongyu Song, Yuchen Liu, Maodi Wu, Haojie Zhang, Bo Jing, Weimin Lai, Xiaobin Gu, Yue Xie, Xuerong Peng, Guangyou Yang

**Affiliations:** 1Department of Parasitology, College of Veterinary Medicine, Sichuan Agricultural University, Chengdu 611130, China; Dongxiaowei1226@outlook.com (X.D.); xujing90@hotmail.com (J.X.); Songhongyu95@outlook.com (H.S.); Liuyuchen1229@163.com (Y.L.); tuzigeyao@163.com (M.W.); jazzachieve@outlook.com (H.Z.); ChenYH1995@hotmail.com (B.J.); Wenruiwei1995@hotmail.com (W.L.); guxiaobin198225@126.com (X.G.); Zhandegaokandey123@163.com (Y.X.); 2Department of Chemistry, College of Life and Basic Science, Sichuan Agricultural University, Chengdu 611130, China; pxuerong@aliyun.com

**Keywords:** *Dirofilaria immitis*, cystatin, immunohistochemistry, peripheral blood mononuclear cells, immune evasion

## Abstract

Infection with canine heartworm (*Dirofilaria immitis*), spread via mosquito vectors, causes coughing, asthma, pneumonia, and bronchitis in humans and other animals. The disease is especially severe and often fatal in dogs and represents a serious threat to public health worldwide. Cysteine protease inhibitors (CPIs), also known as cystatins, are major immunomodulators of the host immune response during nematode infections. Herein, we cloned and expressed the cystatin *Di*-CPI from *D. immitis*. Sequence analysis revealed two specific cystatin-like domains, a Q-x-V-x-G motif, and a SND motif. Phylogenetic analysis indicates that *Di*-CPI is a member of the second subgroup of nematode type II cystatins. Probing of *D. immitis* total proteins with anti-r*Di*-CPI polyclonal antibody revealed a weak signal, and immunofluorescence-based histochemical analysis showed that native *Di*-CPI is mainly localized in the cuticle of male and female worms and the gut of male worms. Treatment of canine peripheral blood mononuclear cells (PMBCs) with recombinant *Di*-CPI induced a Th2-type immune response characterized by high expression of the anti-inflammatory factor interleukin-10. Proliferation assays showed that *Di*-CPI inhibits the proliferation of canine PMBCs by 15%. Together, the results indicate that *Di*-CPI might be related to cellular hyporesponsiveness in dirofilariasis and may help *D. immitis* to evade the host immune system.

## 1. Introduction

The filarial nematode *Dirofilaria immitis*, also known as canine heartworm, mainly occurs in tropical and temperate regions throughout the world, and causes canine and feline cardiopulmonary dirofilariasis as well as human pulmonary dirofilariasis [1]. Females of mosquito species in the Culicidae family, such as *Culex*, *Aedes*, *Anopheles*, and *Culiseta*, are the main vectors of *D. immitis* [2]. Dogs are the most suitable definitive host, which serves as a reservoir. A typical symptom in dogs is a persistent, chronic, and unproductive cough which is aggravated by exercise, and in most cases is accompanied by dyspnea and/or stress tachypnea [3,4]. However, as incidental hosts of *D. immitis*, feline cardiopulmonary dirofilariasis and human pulmonary dirofilariasis are habitually misdiagnosed due to the asymptomatic nature of these diseases [1,5].

Inflammation is the main consequence of this disease [6,7], and arteries are affected initially, followed by the lung parenchyma and right heart chamber [1]. Given that *D. immitis* worms are continuously exposed to the host immune system, and yet can achieve a lifespan of years, they must engage in various immune evasion strategies [8]. During short-term immune evasion, infective larvae avoid host immune responses by releasing surface antigens, whereas in long-term immune evasion, pre-adult and adult worms mask their surface by adsorbing different host molecules and cells [9,10].

Cystatins are native, reversible, tight-binding inhibitors of members of the papain (C1) and legumain (C13) families of cysteine proteases (see the MEROPS database; http://merops.sanger.ac.uk). Based on primary sequence homology, the presence or absence of disulfide bonds, and physiological localization, cystatins are assigned to three subfamilies: stefins (type I cystatins), cystatins (type II cystatins), and kininogens (type III cystatins) [11]. Initially, cystatins were characterized as inhibitors of endogenous cysteine proteases that block the active site through noncovalent binding of a highly conserved Q-x-V-x-G motif [12,13,14]. However, alternative functions for cystatins have since been proposed, including immunoregulation, and this has been investigated for nematode cystatins [11,15,16,17]. For instance, the cystatin cysteine protease inhibitor (CPI)-2 from *Brugia malayi* potentially interferes with antigen processing and presentation by inhibiting host proteases [18,19]. In addition, filarial cystatins hold promise for treating allergic and inflammatory diseases [16,17,20,21,22,23] by regulating cytokine production in different regulatory cell types, including upregulation of interleukin (IL)-10 [17,20,21,23,24,25,26,27] and tumor necrosis factor (TNF)-α [24] and downregulation of IL-4 [20], IL-5, and IL-13 [21]. The main target cell type of cystatins are monocytes/macrophages, both in vitro and in vivo [15,24,25,28].

Although studies on some filarial nematode cystatins have been reported, *D. immitis* proteins remain poorly understood. The aims of the present study were (i) to express and characterize cystatin *Di*-CPI from *D. immitis* (ii), determine the immunogenicity of *Di*-CPI and investigate its tissue distribution in female and male parasites, and (iii) examine the role of *Di*-CPI in the immune system of *D. immitis* through in vitro cell assays.

## 2. Material and Methods

### 2.1. Animals

Two female 8-week-old New Zealand white rabbits (3.0−3.5 kg) were obtained from Dashuo Laboratory Animal Co., Ltd. (Chengdu, China). All animals were housed and fed as previously reported [29].

### 2.2. Parasites and Sera

Adult *D. immitis* worms were obtained from an adult dog that died suddenly, and *D. immitis* serum was isolated from a naturally infected 2-year-old dog (10 kg) diagnosed by PCR in a veterinary hospital in Sichuan province, Chengdu, China. Negative serum was obtained from a healthy 3-month-old dog (4 kg), also from a veterinary hospital in Sichuan province, Chengdu, China.

### 2.3. Cloning and Expression of Di-CPI and Di-TIM

Total RNA was extracted from adult mixed-sex *D. immitis* worms using an RNA extraction kit (Tiangen, Beijing, China) and transcribed into cDNA using a first-strand cDNA synthesis kit (Thermo Scientific, MA, USA) according to the manufacturer’s instructions. The full length coding sequence of *Di*-CPI was amplified from *D. immitis* cDNA using the primers 5′-CCGGAATTCGTCACCGGTATTATGGAA-3′ (forward, *Eco*RI site underlined) and 5′-CCGCTCGAGAATCTCTTATTCATGGTAGTACTTT-3′ (reverse, *Xho*I site underlined), designed based on our assembled and annotated *D. immitis* transcriptome [30], and ligated into the pET-32a (+) expression vector (Novagen, NJ, USA). The resulting recombinant plasmid was transformed into *Escherichia coli* BL21 (DE3) cells (Invitrogen, Carsbad, CA, USA). The gene encoding the control protein triosephosphate isomerase of *D. immitis* (*Di*-TIM) was cloned and expressed as above using the primers 5′-CGCGGATCCATGTCGCGAAAATTTCTTGT-3′ (forward, *Bam*HI site underlined) and 5′-CCCAAGCTTTTAATCACGTGCATGAATAATTT-3′ (reverse, *Hind*III site underlined). Recombinant proteins were expressed following induction with 1 mM isopropyl β-D-thiogalactoside (IPTG) for 6 h, and then purified by Ni^2+^ affinity chromatography (Bio-Rad, Hercules, CA, USA). EndoTrap columns were used to remove endotoxin according to the manufacturer’s instructions (Bioendo, Xiamen, China). Final endotoxin concentrations were measured by Limulus amoebocyte lysate tests (Bioendo, Xiamen, China). The average endotoxin concentration for r*Di*-CPI was 0.025 endotoxin units (EU) per application, and for r*Di*-TIM, it was 0.05 EU per application. The protein concentration after endotoxin removal was measured using a NanoDrop spectrophotometer (Thermo Scientific, MA, USA). Purified recombinant CPI (2 mg/mL) and TIM (1 mg/mL) (with endotoxin removed) were stored at −80°C until needed.

### 2.4. Molecular Characterisation of Di-CPI

The open reading frame (ORF) tool ORFfinder (http://www.ncbi.nlm.nih.gov/orffinder/) was used to analyze the *Di*-CPI ORF and deduce the amino acid sequence, and the SignalP 4.1 Server (http://www.cbs.dtu.dk/services/SignalP/) was used to predict the signal sequence of *Di*-CPI. Molecular weight, isoelectric point, and conserved domains were predicted using ExPASy online software tools (https://www.expasy.org/). In addition, the secondary structure of *Di*-CPI was predicted using YASPIN secondary structure prediction (http://www.ibi.vu.nl/programs/yaspinwww/). Based on similarity, multiple sequence alignment was performed using ClustalX version 2 [31] and a phylogenetic tree was constructed by the maximum likelihood (ML) method using MEGA 6.0 software [32].

### 2.5. Preparation of Polyclonal Antibody Against rDi-CPI

Polyclonal antibody against r*Di*-CPI was produced essentially as previously reported using two female New Zealand rabbits [29]. Briefly, before immunization, rabbit sera were collected to serve as negative controls, and each rabbit was then immunized subcutaneously with 200 μg of r*Di*-CPI emulsified with an equal volume of Freund’s complete adjuvant (Sigma, CA, USA), followed by two boosters every two weeks using the same route and dose, but with Freund’s incomplete adjuvant. At two weeks after the third immunization, sera were collected and purified using HiTrap Protein A affinity chromatography (Bio-Rad), and the resulting IgG was stored at −80°C until needed.

### 2.6. Western Blotting and Immunochemical Analysis 

For immunoblot analysis, *D. immitis* total proteins were obtained using a mammalian protein extraction kit (CWBIO, Beijing, China). Purified r*Di*-CPI and total proteins were separated by 12% sodium dodecyl sulphate-polyacrylamide gel electrophoresis (SDS-PAGE) and transferred onto a nitrocellulose membrane. The rest of the Western blotting procedure was performed as described previously [33].

For immunohistochemistry studies, *D. immitis* sections were incubated with a 1/100 dilution of rabbit anti-r*Di*-CPI serum overnight at 4 °C, and then incubated with fluorescein isothiocyanate (FITC)-conjugated goat anti-rabbit IgG antibody (1:100 dilution in 1% Evans Blue; Bethyl Laboratories, Montgomery, TX, USA) as described previously [30]. Finally, sections were analyzed with a fluorescence microscope (Olympus, Tokyo, Japan).

### 2.7. Cell Isolation and Culturing

For isolation of peripheral blood mononuclear cells (PBMCs) from the blood of a healthy dog, density gradient centrifugation was performed using the Ficoll–Hypaque method (Solarbio, Beijing, China), and samples were washed twice with phosphate-buffered saline (PBS). Isolated PBMCs were resuspended in RPMI 1640 (Hyclone, CA, USA) containing 10% fetal calf serum (PAA, Germany), 100 U/mL penicillin, 100 mg/mL streptomycin, and 20 mmoL/L L-glutamine (Biochrom, Germany), and counted with a Cell Counting Chamber (Qiujing, Shanghai, China).

### 2.8. Proliferation Assay

PBMCs (3.5 × 10^5^/well) were stimulated with 5 μg/mL phytohemagglutinin (PHA) (Solarbio, Beijing, China) and 2 μg/mL IL-2 (Peprotech, CT, USA), and incubated with r*Di*-CPI (1 μg/mL, 2 μg/mL, 5 μg/mL) or r*Di*-TIM (5 μg/mL) for 48 h, then incubated with Cell Counting Kit-8 (CCK-8) (Dojindo, Kyushu, Japan) (10 μL/well) for 4 h. Samples for each group were repeated in triplicate wells. Proliferative responses were analyzed and the optical density at 450 nm was measured using a microplate reader (Thermo Scientific, MA, USA).

### 2.9. Cytokine Analysis

Canine-specific enzyme-linked immunosorbent assay (ELISA) antibody pairs for IL-4, IL-10, IL-12, TNF-α, and IFN-γ (Zhuocai, Shanghai, China) were used according to the manufacturer’s recommendations to determine cytokine levels in cell culture supernatants of PBMCs. Briefly, PBMCs were isolated by density gradient centrifugation and cultured (3.5 × 10^5^ cells/well) in 96-well flat-bottom plates at 37 °C for 6, 24, 48, or 72 h in the presence of 5 μg/mL r*Di*-CPI or 5 μg/ml control protein (r*Di*-TIM). All samples were analyzed in triplicate.

### 2.10. Statistical Analysis 

Multiple groups were compared by one-way analysis of variance (ANOVA) followed by Tukey’s tests, and pairs of groups were compared using Student’s *t*-tests or Mann–Whitney U-tests using SPSS version 20.0 (SPSS Inc., USA). Data are presented as mean values ± standard error of the mean (SEM), determined using Prism 6 (Graphpad Software Inc., CA, USA). Values of *p* < 0.05 were considered statistically significant.

### 2.11. Ethics Statement

All animals were handled in strict accordance with the Animal Protection Law of the People’s Republic of China (released on 18 September 2009). All procedures were carried out in line with the Regulations for Care and Use of Laboratory Animals of the Animal Ethics Committee of Sichuan Agricultural University (Ya’an, China; Approval No. 2015–028).

## 3. Results

### 3.1. Molecular Characterisation of Di-CPI

The full-length *Di*-CPI sequence was amplified from mixed-sex adult *D. immitis* cDNA samples and confirmed to be identical to the sequence obtained from the annotated *D. immitis* transcriptome [31]. The *Di*-CPI protein consists of 125 amino acid residues, has a predicted molecular weight of 14.5 kDa, and has a calculated isoelectric point (pI) of 5.45. An N-terminal signal sequence was not found. Sequence analysis revealed high sequence identity with orthologs from *Onchocerca volvulus* (85%), *Brugia malayi* (82%), *Litomosoides sigmodontis* (79%), *Loa loa* (75%), and *Acanthocheilonema viteae* (72%). Despite low overall sequence identity (20–30%) between *Di*-CPI and vertebrate type II cystatins (*Mus musculus* = 26%, *Homo sapiens* = 30%, *Canis lupus familiaris* = 25%), all the key structural features are conserved, including the conserved inhibitory domain signature Q-x-V-x-G associated with papain inhibition, the N-terminal glycine residue, the C-terminal PW motif, and a single disulfide bond. In addition, a SND motif associated with asparaginyl endopeptidase (AEP) inhibition was observed (Figure 1).

To investigate the evolutionary history of *Di*-CPI, amino acid sequences of both mammalian and nematode cystatins were aligned and subjected to phylogenetic analysis using an ML tree (Figure 2). Combining the results of sequence analysis with phylogenetic analysis, *Di*-CPI appears to be a member of the second subgroup of nematode type II cystatins.

### 3.2. Expression and Characterisation of Di-CPI

The *Di*-CPI cDNA was successfully inserted into the pET32a (+) expression vector, and the protein was expressed in *E. coli* BL21 (DE3) cells in soluble form. The recombinant protein was ~35 kDa, according to SDS-PAGE, and contains a ~20 kDa epitope tag (including Trx-tag, His-tag, and S-tag) fusion protein (Figure 3, lane 1). Purified protein was prepared for Western blotting and immunohistochemistry analyses (Figure 3, lane 2). Western blotting yielded a single band at ~35 kDa that was recognized by dog anti-*D. immitis* serum (Figure 3, lane 3). Additionally, a weak band at ~15 kDa was observed when total protein from *D. immitis* was probed with anti-r*Di*-CPI polyclonal antibody (Figure 3, lane 5), which corresponds to the predicted size of native *Di*-CPI.

The distribution of *Di*-CPI in transverse sections of adult *D. immitis* worms was assessed by immunofluorescence using rabbit anti-r*Di*-CPI antibody. Immunohistochemical analysis revealed that native *Di*-CPI was mainly distributed in cuticle of both male and female worms (Figure 4A,B), and a large amount of fluorescence was also observed in the gut of male worms (Figure 4B).

### 3.3. Inhibition of the Proliferation of Canine PBMCs by rDi-CPI

To determine whether *Di*-CPI is involved in host cellular responses, we firstly evaluated the effect of r*Di*-CPI on T-cell proliferation induced by IL-2 and PHA. At 48 h post-treatment, 5 μg/mL of r*Di*-CPI inhibited the proliferation of canine PBMCs by 15% (Figure 5, *p* < 0.01). Compared with the control protein, r*Di*-TIM, r*Di*-CPI significantly inhibited proliferation by 10% (*p* < 0.05). There was no significant difference in T-cell proliferation among samples treated with 1 μg/mL, 2 μg/mL, or 5 μg/mL of r*Di*-CPI; one of the possible explanations for this is that the lowest amount of r*Di*-CPI was already above saturation.

### 3.4. Cytokine Analysis

Based on our preliminary data, we hypothesized that inhibition of T-cell proliferation may be associated with the production of cytokines. In order to investigate *Di*-CPI-induced immune responses, we quantified the levels of both Type 1 T helper (Th1) and Type 2 T helper (Th2) cytokines in PBMCs. Following incubation for 6 h, r*Di*-CPI had induced the production of IL-10, and the peak in IL-10 production (Figure 6A, *p* < 0.01) was followed by a significant decrease of the production of IL-12 (Figure 6B, *p* < 0.01) and TNF-α (Figure 6C, *p* < 0.05) at 48 h. However, no significant production of IFN-γ and IL-4 was observed with r*Di*-CPI after 48 h compared with controls (Figure 6D,E). 

## 4. Discussion

In the present study, we cloned, expressed, and characterized *Di*-CPI from *D. immitis* based on the transcriptome sequence [30]. Based on both sequence similarity and phylogenetic analyses, *Di*-CPI is clearly a member of the cystatin superfamily and most likely a member of the second subgroup of type II nematode cystatins. Type II cystatins from nematodes fall into three subgroups, represented by *Bm*-CPI-1, *Bm*-CPI-2, and *Bm*-CPI-3 in *B. malayi* [34]; *Ov*-CPI-1 and *Ov*-CPI-2 in *O. volvulus* [24,35,36]; and *Ce*-CPI-1 and *Ce*-CPI-2 in *C. elegans* [18,37]. Among the three subgroups, the second subgroup that includes *Bm*-CPI-2 is perhaps the most intriguing. In addition to the classical inhibitory that blocks C1 papain-like proteases such as cathepsins B, L, and S, the second subgroup of filarial cystatins also possess a site resembling a feature in the vertebrate cystatin C protein that controls C13 legumain-like asparaginyl endopeptidase (AEP) [35,38,39]. The SND motif in mammalian cystatin C, which is also present in *Bm*-CPI-2, is associated with AEP inhibitory activity and suppresses the activity of mammalian AEP [18,19]. Intriguingly, sequence alignment revealed that *Di*-CPI has an SND motif, which is also present in other second subgroup filarial cystatins such as those from *A. viteae* [25], *L. sigmodontis* [40], and *O. volvulus* [36] (Figure 1). Hence, we speculate that *Di*-CPI may have the potential to block mammalian AEP activity, but further experiments are required to confirm this hypothesis.

To date, research on the location of native cystatin in *D. immitis* has not been reported. It is intriguing to note that this protein is expressed in the cuticle of both male and female adult *D. immitis* worms, and also in the gut of male worms. According to a previous study, both cystatin (*Di*-CPI) and triosephosphate isomerase (*Di*-TIM, or *Di*-TPI) are among the most abundant proteins secreted into a culture medium by *D. immitis* [41]; however, neither of these proteins have an apparent signal sequence. In recent studies, parasites have been shown to release exosome-like vesicles [42,43,44,45,46,47] which function as cell-to-cell effectors in the host–parasite interaction. Therefore, one of the possible explanations is that *Di*-CPI was released to the extracellular environment as the cargo of exosomes by fusion of multivesicular bodies with the plasma membrane. Confirmation of this speculation requires further study. Adult *D. immitis* worms are believed to produce numerous proteins to mask their surface and thereby avoid host immune responses [10]. Similarly, surface-expressed CPIs might help *D. immitis* evade host immune responses. *Bm*-CPI-2 mRNA was detected throughout all developmental stages of the life cycle in the mosquito vector and mammalian hosts, indicating a positive role in parasite maintenance [19]. Additionally, a homologous gene product is highly expressed in the cuticle of molting larvae in *O. volvulus* [48]. In the present study, we only probed the distribution of *Di*-CPI in adult worms, and further investigation of expression levels during each developmental stage is clearly needed. We speculate that *Di*-CPI may not only be active in adult worms, but also larvae, and may be associated with maturation and molting in *D. immitis*, but further experiments are required for confirmation.

In the immunopathogenic mechanisms of *D. immitis*, a dual Th1/Th2 response was observed in BALA/c mice immunized with antigenic extracts from *D. immitis* adult worms (DiSA) [1,49]. The Th2-type anti-inflammatory response was stimulated mainly by *D. immitis* antigens, whereas the Th1-type proinflammatory response was stimulated by *Wolbachia* bacteria released from dying worms [49]. The Th2-type response was characterized by the high levels of IL-4 and IL-10 mRNAs, and IgG1 production was predominant in microfilaremic canine infections [50]. In the present work, we simultaneously measured the expression of IL-4 and IL-10 in *Di*-CPI-induced canine PBMCs and found that *Di*-CPI induced a Th2-type anti-inflammatory immune response characterized by high expression of IL-10. High levels of IL-10 are characteristic of filarial infections and are linked to hyporesponsiveness in lymphatic filariasis [25,51,52]. Furthermore, IL-10 induced by cystatins is reportedly effective against dextran sodium sulfate (DSS)-induced intestinal inflammation [27], mucosal inflammation [53], and colitis [26]. In addition, *Di*-CPI also suppressed the Th1-type proinflammatory immune response by reducing the production of IL-12 and TNF-α. We also found that *Di*-CPI inhibited canine T-cell proliferation by 15%, and hypothesized that *Di*-CPI participates in immune evasion in canine heartworm disease. Evidence suggests that filarial cystatins are involved in immune interference, and they may be important immune evasion molecules [11,54,55,56]. For instance, *A. viteae* CPI-2 (Av17) directly inhibits the proliferation of murine T cells in vitro and induces the production of cytokine IL-10 [25]. Meanwhile, *O. volvulus* CPI-2, also known as OV-17 and onchocystatin, significantly inhibits human T-cell proliferation [28]. Thus, high expression of IL-10 induced by *Di*-CPI, together with effective inhibition of canine T-cell proliferation by *Di*-CPI, indicates that *Di*-CPI plays a positive role in immune evasion and may be an effective inhibitor of host inflammatory responses. It is worth noting that changes in cytokine levels varied from study to study [17,21,24,25,57]. In the present work, we observed numerically modest cytokine level changes; this is probably due to the different parasite cystatins and different host PBMCs, as PBMCs from different hosts may show varied responses to a specific parasite cystatin and vice versa.

## 5. Conclusions

In summary, we characterized the cystatin *Di*-CPI from *D. immitis* and observed high expression on the surface of male and female adult worms following immunolocalization analysis. Additionally, we demonstrated its immune regulation ability using ex vivo canine PBMCs. The findings show that *Di*-CPI inhibits canine T-cell proliferation and modulates inflammatory responses by inducing expression of IL-10 and reducing expression of IL-12 and TNF-α. Thus, *Di*-CPI may be an important immune evasion molecule and may play an important role in host–parasite interactions.

## Figures and Tables

**Figure 1 genes-10-00300-f001:**
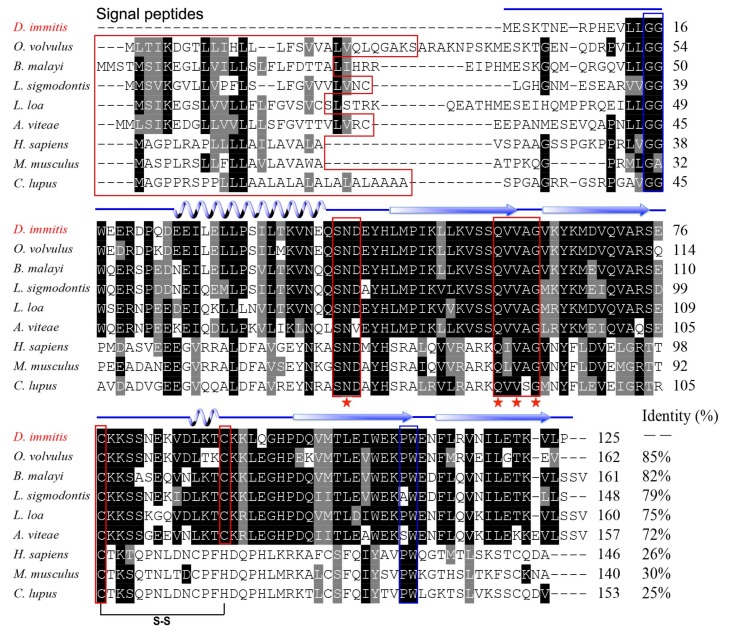
Sequence alignment of *Dirofilaria immitis* cysteine protease inhibitor (*Di*-CPI). Alignment of the deduced *Di*-CPI amino acid sequence with those of homologous proteins from nematodes (*Onchocerca volvulus*, AAA29423.1; *Brugia malayi*, AAD51086.1; *Litomosoides sigmodontis*, AAF35896.1; *Loa loa*, XP_003147913; *Acanthocheilonema viteae*, AAA87228.1) and mammals (*Mus musculus*, NP_034106.2; Human, AAH13083.1; Canine, XP_003639869.1) was performed using Clustal X software version 2 and shaded using BOXshade version 3.21. Predicted secondary structural elements, including  strands and helices, are shown above the alignment as arrows and loops, respectively. The Q-x-V-x-G motif and SND motif are enclosed in red boxes. A conserved glycine residue in the N-terminal region and the PW hairpin loop in the C-terminal region are enclosed in the blue boxes. N-terminal signal peptides are enclosed in a red box.

**Figure 2 genes-10-00300-f002:**
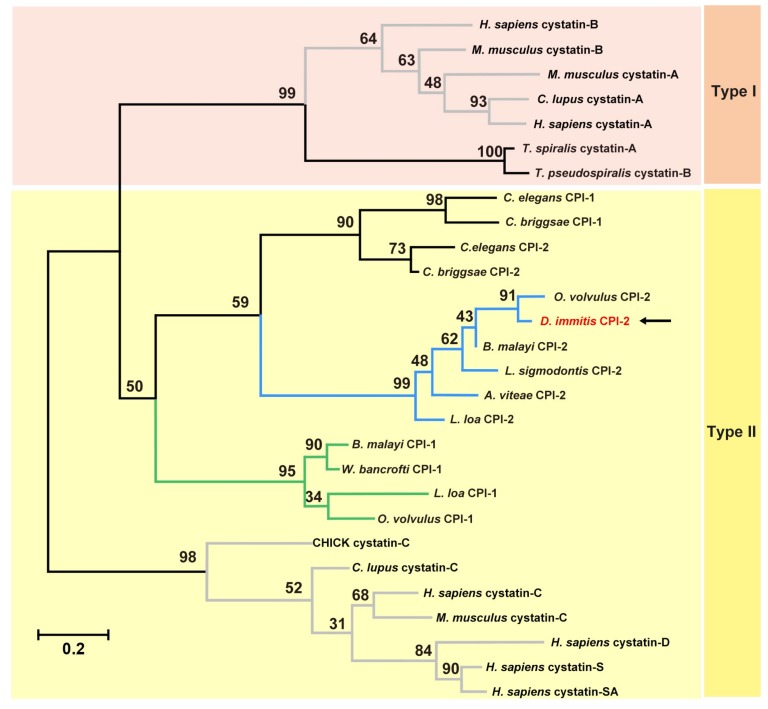
Phylogenetic relationships between *Di*-CPI and homologous cystatins/CPIs. The tree was constructed from a multiple sequence alignment performed using Clustal W2 and plotted using MEGA 6.06. Amino acid sequences used in the tree, with their GenBank accession numbers, are as follows: *Homo sapiens* cystatin-B, NP_000091.1; *Mus musculus* cystatin-B, NP_001076014.1; *Mus musculus* cystatin-A, NP_005204.1; *Canis lupus familiaris* cystatin-A, SPO16019.1; *Homo sapiens* cystatin-A, NP_005204.1; *Trichinella spiralis*, KRY35881.1; *Trichinella pseudospira*, KRZ14801.1; *Caenorhabditis elegans* CPI-1, NP_500915.1; *Caenorhabditis briggsae* CPI-1, CAP25989.2; *Caenorhabditis elegans* CPI-2, AF068718.1; *Caenorhabditis briggsae* CPI-2, CAP29292.1; *Onchocerca volvulus* CPI-2, |P22085.2; *Brugia malayi*, AAD51086.1; *Litomosoides sigmodontis*, AAF35896.1; *Acanthocheilonema viteae*, AAA87228.1; *Loa loa* CPI-2, XP_003147913.1; *Loa loa* CPI-1, XP_003136654.1; *Brugia malayi* CPI-2, U80972.1; *Wuchereria bancrofti*, EJW82673.1; *Onchocerca volvulus* CPI-1, AF177194.1; CHICK, P01038.2; *Canis lupus familiaris* cystatin-C, XP_003639869.1; *Homo sapiens* cystatin-C, P01034.1; *Homo sapiens* cystatin-D, P28325.1; *Homo sapiens* cystatin-S, P01036.3; *Homo sapiens* cystatin- SA, P09228.1. Numbers indicate bootstrap values. The branches in grey and in black represent mammalian cystatins and nematode cystatins, respectively. In addition, the green and blue branches represent first subgroup of filarial cystatins and the second subgroup of filarial cystatins, respectively. The pink and yellow backgrounds represent type I and type II cystatins, respectively.

**Figure 3 genes-10-00300-f003:**
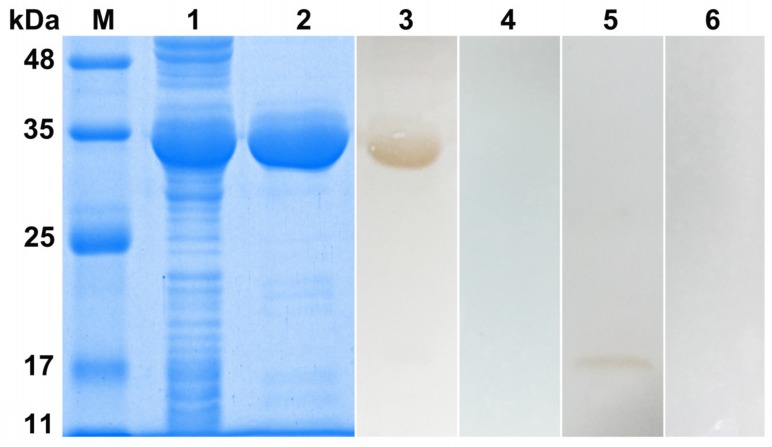
Sodium dodecyl sulphate–polyacrylamide gel electrophoresis (SDS-PAGE) and Western blotting analysis of *Di*-CPI. M, molecular weight markers; Lane 1, IPTG-induced recombinant (r) *Di*-CPI in *Escherichia coli* BL21 (DE3); Lane 2, purified r*Di*-CPI (6 μg); Lane 3, purified r*Di*-CPI probed with serum from a dog naturally infected with *D. immitis* (6 μg); Lane 4, purified r*Di*-CPI probed with naive dog serum; Lane 5, total protein from *D. immitis* probed with anti-r*Di*-CPI serum (20 μg); Lane 6, total protein from *D. immitis* probed with native (preimmune) rabbit serum (20 μg).

**Figure 4 genes-10-00300-f004:**
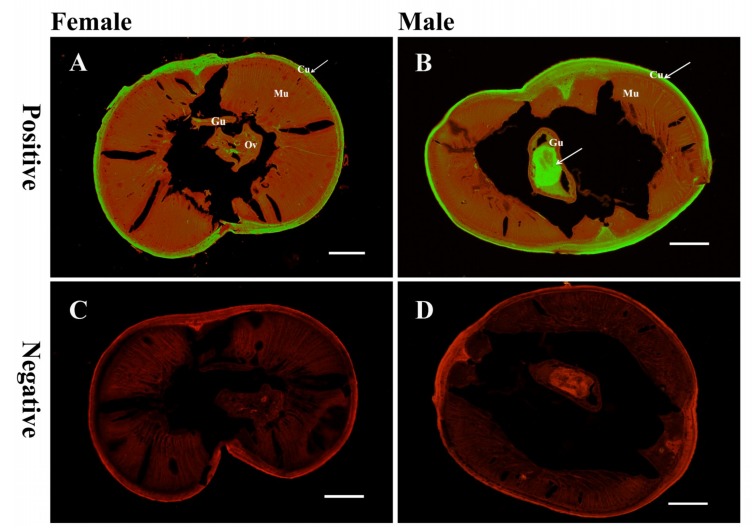
Immunolocalization of native *Di*-CPI. Both adult female and male worms were probed. The location of native *Di*-CPI protein is indicated by green fluorescence. Sections (5 μm) were incubated with either rabbit anti-r*Di*-CPI IgG at 1:100 (**A** and **B**) or preimmune IgG at 1:100 (**C** and **D**), diluted in phosphate-buffered saline (PBS). Arrows indicate a positive signal in the cuticle of female worms (A) and male worms (B) and the gut of male worms (B). No staining was observed for female or male worms using IgG purified from preimmune serum (C and D), confirming specific immunolabeling was achieved. Cu, cuticle; Gu, gut; Ov, ovary. Scale bars = 100 μm.

**Figure 5 genes-10-00300-f005:**
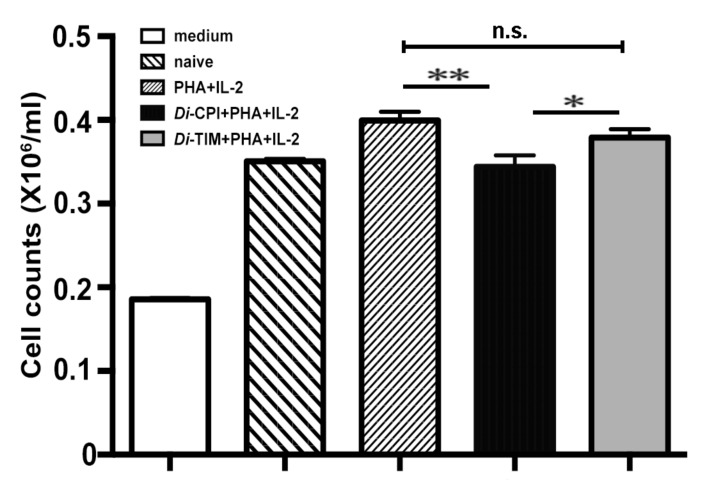
Inhibition on the proliferation of canine peripheral blood mononuclear cells (PBMCs) by *Di*-CPI. Canine T cells were stimulated with PHA + IL-2, PHA + IL-2 + r*Di*-CPI, or PHA + IL-2 + r*Di-*TIM for 48 h as determined by CCK-8. *Di*-CPI inhibited the proliferation of canine T cells by 15% compared with naive controls and 10% compared with *Di*-TIM controls. Naive indicates no protein was added to wells. Data are mean ± SEM of 3 samples with triplicate wells; representative data from one of three independent experiments. Statistically significant differences (* *p* < 0.05, ** *p* < 0.01) were determined by Student’s *t*-tests.

**Figure 6 genes-10-00300-f006:**
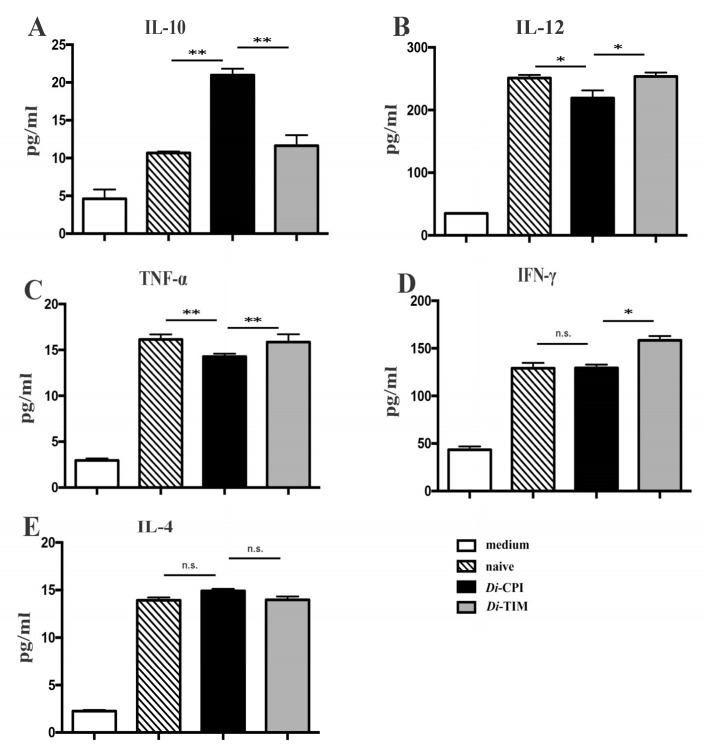
Analysis of cytokines following inhibition of canine PBMCs by *Di*-CPI. Supernatants of canine PBMCs stimulated for 48 h were subjected to ELISA to measure the concentration of IL-10, IL-12, TNF-α, IFN-γ, and IL-4. r*Di*-CPI treatment increased anti-inflammatory cytokine IL-10 (**A**), followed by significant decrease of the production of IL-12 (**B**) and TNF-α (**C**). IFN-γ (**D**) and IL-4 (**E**) showed no significant difference compared with controls. Naive indicates no protein was added to wells. Data are mean ± SEM of 3 samples with triplicate wells; representative data from one of three independent experiments. n.s., not significant. Statistically significant differences (* *p* < 0.05, ** *p* < 0.01) were determined by Mann–Whitney U-tests.

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
