# Peer review of "Molecular Characterization of a Dirofilaria immitis Cysteine Protease Inhibitor (Cystatin) and Its Possible Role in Filarial Immune Evasion"

_genes, 2019, doi:10.3390/genes10040300_

Round 1

Reviewer 1 Report

This manuscript presents interesting data on a cystatin from the filarial parasite Dirofilaria immitis. The manuscript is concise and well-written, but improvement is necessary before it should be published.

The authors stress the potential importance of human dirofilariosis, but this is a very small problem compared to the infection in dogs and cats, and should not be emphasized.

The authors neglect to mention that both cystatin and TPI are among the most abundant proteins secreted into culture medium by D. immitis (Geary et al., 2012, Parasites & Vectors).  The lack of an apparent signal sequence suggests that cystatin may be secreted in exosomes.  The possible implications of secretion should be discussed vis-a-vis the localization data provided here.

The choice of a single concentration of cystatin for biologic evaluation needs to be justified.  Is 0.5 uM a biologically relevant concentration (7.5 ug/ml)?

The effects on cytokine levels are numerically very modest; are such small changes biologically relevant?  The authors should provide comparative data for other cytokine modulators, preferably other cystitis, to indicate relative potency and efficacy of the heartworm protein. 

It is important to know if a single independent biological replicate was run (with 3 samples) for the cell assays.  It is essential run at least 3 independent biological replicates for these assays, each with triplicate wells (n = 9).

Clarify that "naive" means no protein added to wells.

Author Response

General comments:

This manuscript presents interesting data on a cystatin from the filarial parasite Dirofilaria immitis. The manuscript is concise and well-written, but improvement is necessary before it should be published.

Response: Thank you very much for your constructive comments and suggestions on our manuscript. Your comments and suggestions are very professional and valuable, and have been taken very seriously. We have revised the paper accordingly, and detailed point-by-point responses to these comments and/or suggestions are provided below.

Specific Critiques:

Point 1: The authors stress the potential importance of human dirofilariosis, but this is a very small problem compared to the infection in dogs and cats, and should not be emphasized.

Response: Thanks for your kind suggestions. We have revised our description in Introduction. (see lines: 38, 40-43)

Point 2: The authors neglect to mention that both cystatin and TPI are among the most abundant proteins secreted into culture medium by D. immitis (Geary et al., 2012, Parasites & Vectors).  The lack of an apparent signal sequence suggests that cystatin may be secreted in exosomes.  The possible implications of secretion should be discussed vis-a-vis the localization data provided here.

Response: Thanks for your professional comments, that’s very helpful. We have added corresponding contents to the discussion (see lines: 297-304).

Point 3: The choice of a single concentration of cystatin for biologic evaluation needs to be justified.  Is 0.5 uM a biologically relevant concentration (7.5 ug/ml)?

Response: Thank you for your reminding. Actually, we used three concentrations of cystatin (1mg/ml, 2mg/ml, 5mg/ml) for proliferation assay. There was no significant difference in T cell proliferation among samples treated with 1mg/ml, 2mg/ml, or  5mg/ml rDi-CPI, so we only chose one concentration (5mg/ml) for cytokine analysis. In our experiment, 0.5mM is equivalent to 5mg/ml. We have justified our description and corrected the unit of the concentration. (see lines: 143-144, 152-153, 244-245, 247-248)

Point 4: The effects on cytokine levels are numerically very modest; are such small changes biologically relevant?  The authors should provide comparative data for other cytokine modulators, preferably other cystitis, to indicate relative potency and efficacy of the heartworm protein. 

Response: Thank you for your constructive comments! We have revised our manuscript (see lines: 337-340).

Point 5: It is important to know if a single independent biological replicate was run (with 3 samples) for the cell assays.  It is essential run at least 3 independent biological replicates for these assays, each with triplicate wells (n = 9).

Response: Thank you, as you suggested, replicates are really important for almost every experiments, this is the very basic knowledge when we design our experiments (you can see our published paper) [1-3]. Actually, in this study, we do more than three times for each experiment to make sure the results are repeatable and reliable. As you can see, effects of the cytokine are numerically modest in this study; therefore, we must make sure our results can be trusted. We have specified these details in our revised manuscript. (see lines: 254-256, 275-276)

1. Xie Y, Chen S, Yan Y, Zhang Z, Li D, et al. (2013) Potential of recombinant inorganic pyrophosphatase antigen as a new vaccine candidate against Baylisascaris schroederi in mice. 44: 1-16.

2. Song X, Min Y, Hu D, Yu W, Ning W, et al. (2016) Molecular characterization and serodiagnostic potential of a novel dithiol glutaredoxin 1 from Echinococcus granulosus. 9: 456.

3. Xu J, Huang X, Dong X, Ren Y, Wu M, et al. (2018) Serodiagnostic Potential of Alpha-Enolase From Sarcoptes scabiei and Its Possible Role in Host-Mite Interactions. 9: 1024.

Point 6: Clarify that "naive" means no protein added to wells.

Response: Thank you. We have been clarified that “naive” means no protein added to wells. (see lines: 254 and 275)

Reviewer 2 Report

Well done study, The results are presented clearly, the conclusions are sound.

Author Response

General comments for authors:

Well done study, the results are presented clearly, the conclusions are sound.

Response: Thank you.

Reviewer 3 Report

The importance of protease and protease inhibitors in relationships among biological species, predators or invaders and attacked or parasitised victims, in defence and immunological tolerance or evasion, among others, is becoming increasingly documented and of great fundamental and applied interest.  The present report deals with this issue, in an atractive, clear-cut and elegantly made way.  

A few suggestions for authors to improve the manuscript:

     -As indicated in Results (line 214), the recombinant expression of Di-CPI yielded a band of about 35KDa, which is supposed to be a fusion protein of the cystatin+a 20KDa epitope tag.  it would be useful to explicitly indicate which kind of epitope tag is this one and place it also in a visible site of Materials an Methods or Results.  Also, provide information about the production yield of the recombinant protein expression and purification of Di-CPI, which is essential for both the generation of antibodies as well as for the proliferation and cytokine analyses.

     -As it is explained in the manuscript, it would seem that authors detected and fished Di-CPI directly from transcriptome analysis, by homology analyses with other cystatins.  Is this the reality or they also detected and/or analysed part of the sequence of Di-CPI from parasite extracts (i.e. by proteomics) to facilitate the capture, amplification and cloning of its cDNA?  This aspect should be clarified.

     -In line 229 of Results it is said that "...native Di-CPI was not detected in female worms gut due to incomplete intestinal structure ...".    What is the meaning this?   (incomplete intestinal structure?).  This sentence should be perhaps modified to clarify it.

     -In Results, lines 258 and 266, the last one at Figure 6, it is said that rDi-CPI treatment produced a reduction in the production of IL12 and TNF-alfa.  However, in such Fig.6, parts B and D, it seems that both cytokines apparently suffer a level increase and not a reduction, regarding controls, from a non-expert view (as is this reviewer, in this issue, and most of the expected readers).  Probably the rationale behind of such conclusions is correct but it would be fair that the authors modify the text to facilitate general understanding. 

     -In page 10, line 298, the authors use the term "noteworthiness", which seems to be a correct one (and is in the english dictionary) but the way to include it in the sentence seems to be "strange" or "unusual".  Take it into account, please.

     -In the main title of the manuscript the authors emphasise the role of Di-cystatin in filarial immune invasion.  This is a role already proposed by other authors in the field of parasites, and it seems that the cellular cultures-based experiments done in this work add further indications about but, in this reviewer's opinion, such experiments are not sufficient to clearly prove such property as a main one of the D. immitis parasite.  Therefore, as a suggestion for safer conclusions, it would be wise to add the term "potential" or equivalent before "role" in the main title (i.e., "... ,cystatin, and its potential role in filarial immune evasion.") 

Author Response

General comments:

The importance of protease and protease inhibitors in relationships among biological species, predators or invaders and attacked or parasitised victims, in defence and immunological tolerance or evasion, among others, is becoming increasingly documented and of great fundamental and applied interest.  The present report deals with this issue, in an atractive, clear-cut and elegantly made way.  

Response: Thank you very much for your favorite on our manuscript and the constructive comments and suggestions. We have revised the paper accordingly, and detailed point-by-point responses to these comments and/or suggestions are provided below.

Specific Critiques:

A few suggestions for authors to improve the manuscript:

Point 1: -As indicated in Results (line 214), the recombinant expression of Di-CPI yielded a band of about 35KDa, which is supposed to be a fusion protein of the cystatin+a 20KDa epitope tag.  it would be useful to explicitly indicate which kind of epitope tag is this one and place it also in a visible site of Materials an Methods or Results.  Also, provide information about the production yield of the recombinant protein expression and purification of Di-CPI, which is essential for both the generation of antibodies as well as for the proliferation and cytokine analyses.

Response: Thank you for your reminding. The epitope tag type has been added in results (see lines 216-217). The production yield of purified recombinant protein has been marked (see lines 102). The concentration of Di-CPI for cytokine analyses has been corrected (see lines: 143-144, 152-153, 244-245, 247-248).

Point 2: -As it is explained in the manuscript, it would seem that authors detected and fished Di-CPI directly from transcriptome analysis, by homology analyses with other cystatins.  Is this the reality or they also detected and/or analysed part of the sequence of Di-CPI from parasite extracts (i.e. by proteomics) to facilitate the capture, amplification and cloning of its cDNA?  This aspect should be clarified.

Response: Thanks. Actually, we obtained this sequence from the transcriptome data of adult D.immitis that was sequenced in our laboratory before (Fu et al., Plos one, 2012). We have clarified this in our revised manuscript. (see lines: 84-89)

Point 3: -In line 229 of Results it is said that "...native Di-CPI was not detected in female worms gut due to incomplete intestinal structure ...".    What is the meaning this?   (incomplete intestinal structure?).  This sentence should be perhaps modified to clarify it.

Response: Thank you very much. We have deleted this confusion sentence. (see lines: 232-233)

Point 4: -In Results, lines 258 and 266, the last one at Figure 6, it is said that rDi-CPI treatment produced a reduction in the production of IL12 and TNF-alfa.  However, in such Fig.6, parts B and D, it seems that both cytokines apparently suffer a level increase and not a reduction, regarding controls, from a non-expert view (as is this reviewer, in this issue, and most of the expected readers).  Probably the rationale behind of such conclusions is correct but it would be fair that the authors modify the text to facilitate general understanding. 

Response: Thank you very much for reminding us of that, and we shouldn’t make such a mistake. Actually, we measured the cytokine levels in 6h, 24h, 48h and 72h, respectively. Each of the cytokine change was totally different, and we chose the most significant data for drawing. We have renewed the last version of Figure 6 (see line: 269), and revised the relevant results and Figure note (see lines: 260-267, 272-276).

Point 5: -In page 10, line 298, the authors use the term "noteworthiness", which seems to be a correct one (and is in the english dictionary) but the way to include it in the sentence seems to be "strange" or "unusual".  Take it into account, please.

Response: Thank you very much, we have deleted the relevant description. (see lines: 313-314)

Point 6: -In the main title of the manuscript the authors emphasise the role of Di-cystatin in filarial immune invasion.  This is a role already proposed by other authors in the field of parasites, and it seems that the cellular cultures-based experiments done in this work add further indications about but, in this reviewer's opinion, such experiments are not sufficient to clearly prove such property as a main one of the D. immitis parasite.  Therefore, as a suggestion for safer conclusions, it would be wise to add the term "potential" or equivalent before "role" in the main title (i.e., "... ,cystatin, and its potential role in filarial immune evasion.") 

Response: Thank you for your suggestions. Indeed, this title is not appropriate and has been revised. (see line: 3)

Round 2

Reviewer 1 Report

The authors have revised the manuscript in response to my concerns and I believe the new version is acceptable for publication.

Author Response

Thank you very much.